# The immunoglobulin G antibody response to malaria merozoite antigens in asymptomatic children co-infected with malaria and intestinal parasites

**Crespo'o Mbe-cho Ndiabamoh**[1,2]*, **Gabriel Loni Ekali**[2,3], **Livo Esemu**[2,4], **Yukie Michelle Lloyd**[5], **Jean Claude Djontu**[2], **Wilfred Mbacham**[1,2,4], **Jude Bigoga**[2,4], **Diane Wallace Taylor**[5], **Rose Gana Fomban Leke**[2]*

1 Faculty of Medicine and Biomedical Sciences, University of Yaoundé I, Yaoundé, Cameroon, 2 The Biotechnology Center, Nkolbisson, University of Yaoundé I, Yaoundé, Cameroon, 3 National AIDS Control Committee, Ministry of Public Health, Yaoundé, Cameroon, 4 Department of Biochemistry, University of Yaoundé 1, Yaoundé, Cameroon, 5 John A. Burns School of Medicine, University of Hawai'i, Mānoa, Hawaii, United States of America

\* cndiabamoh@gmail.com (CMN); roseleke@yahoo.com (RGFL)

**Data Availability Statement:** All relevant data are within the paper and its Supporting Information files.

## Abstract

### Background

Co-infection with malaria and intestinal parasites is common in children in Africa and may affect their immune response to a malaria parasite infection. Prior studies suggest that co-infections may lead to increased susceptibility to malaria infection and disease severity; however, other studies have shown the reverse. Knowledge on how co-morbidities specifically affect the immune response to malaria antigens is limited. Therefore, this study sought to determine the prevalence of co-infection of malaria and intestinal parasites and its association with antibody levels to malaria merozoite antigens.

### Methods

A cross sectional study was carried out in two villages with high transmission of malaria in Cameroon (Ngali II and Mfou) where mass drug administration (MDA) had been administered at ~6-month intervals (generally with albendazole or mebendazole). Children aged 1–15 years were enrolled after obtaining parental consent. A malaria rapid diagnostic test was used on site. Four (4) ml of peripheral blood was collected from each participant to determine *Plasmodium falciparum* infections by microscopy, haemoglobin levels and serology. Fresh stool samples were collected and examined by wet mount, Kato-Katz method and modified Ritchie concentration techniques. A Multiplex Analyte Platform assay was used to measure antibody levels.

### Results

A total of 320 children were enrolled. The prevalence of malaria by blood smear was 76.3% (244/320) and prevalence of malaria and intestinal parasites was 16.9% (54/320). Malaria prevalence was highest in young children; whereas, intestinal parasites (IP+) were not

**Funding:** The Luminex MAGpix was provided by grant P30GM11473, Centers of Biomedical Research Excellence, National Institute of General Medical Sciences, NIH. GFLR provided the funds for the field work and analysis. The funders had no role in study design, data collection and analysis, decision to publish, or preparation of the manuscript.

**Competing interests:** The authors have declared that no competing interests exist.

present until after 3 years of age. All children positive for malaria had antibodies to $MSP1_{42}$, MSP2, MSP3 and EBA175. No difference in antibody levels in children with malaria-co infections compared to malaria alone were found, except for antibody levels to EBA-175 were higher in children co-infected with intestinal protozoa (p = 0.018), especially those with *Entamoeba histolytica* infections (p = 0.0026).

## Conclusion

Antibody levels to EBA175 were significantly higher in children co-infected with malaria and *E. histolytica* compared to children infected with malaria alone. It is important to further investigate why and how the presence of these protozoans might modulate the immune response to malaria antigens.

## Introduction

In sub-Saharan Africa, malaria caused by *Plasmodium falciparum* (Pf) remains an important public health threat, killing over 271,000 children under the age of five each year [1]. In malaria endemic areas, individuals exposed to malaria infections gradually develop clinical immunity [2] and commonly experience asymptomatic infections without fever or symptoms and do not require antimalarial treatment. Asymptomatic infection results from partial immunity that controls, but does not completely eliminate, malaria parasites, thus allowing for constant presence of circulating parasites [2].

The prevalence of intestinal parasitic infections in children is fairly constant across sub-Saharan Africa with an average prevalence of 26% [3,4]. In Cameroon, the prevalence in children less than 18 years is 26.8% [5], while that for the general population is more than 28% The major intestinal parasites are *Ascaris lumbricoides*, *T. trichiura* and *Entamoeba histolytica* complex [6–8], but many cases of intestinal parasites go undetected.

Co-infections with malaria and intestinal parasites (IP) are common in malaria endemic areas in sub-Saharan Africa [7,8] and infections with IP and Pf are both ranked among the major cause of mortality and morbidity in sub-Saharan Africa. Several studies conducted on IP (intestinal helminths) and Pf have shown conflicting results. Some helminths suppress different T-helper types and favor an increase in regulatory T (Treg) cell [9]. Studies on concomitant infections in humans suggest that *A. lumbricoides* infection may protect against cerebral malaria [10,11], while other studies suggest that children infected by *Schistosoma mansoni may be* more susceptible to *P. falciparum* infections and develop acute malaria episodes [12,13]. Also, it has been shown that the levels of TNF-α, IL-2, IL-10, IL-6 in *Plasmodium*-helminth co-infected individuals were significantly higher than the malaria-positive (MP) group [14] dampening the immune response to malaria. However, to our knowledge, no studies have been conducted regarding host immune responses to malaria in children co-infected with protozoan pathogens.

A previous study suggests that children co-infected with malaria and intestinal helminths had significantly decreased antibody levels to the malarial antigen apical merozoite antigen 1 (AMA-1) compared to those with *P. falciparum* or IP alone [15]. Hence, infections with intestinal helminths can stifle protective anti-plasmodial antibody responses [15]. However, increase in MSP3 IgG1–4 levels were significantly associated with children infected with malaria alone compared to children co-infected with both parasites [15].

Malaria and other intestinal parasites overlap extensively in their epidemiological distributions causing polyparasitism. Polyparasitism with intestinal parasites has been reported as one

of the contributing factors to hypo-responsiveness [16], dampening of the immune response by inducing a strong Treg response, which could in turn, blunt a strong response to vaccines [17]. Equally, some studies have suggested an effect of IP on antibody responses to *P. falciparum* gametocyte antigens that may have consequences on transmission-blocking immunity [18].

Effective elimination and future eradication of malaria will require not only vector control, but also managing asymptomatic malaria patients and developing an effective vaccine. Given the high burden and concomitant nature of both malaria and intestinal parasites in the same geographical setting, it has been suggested that polyparasitism might interfere with the efficacy of future malaria vaccines [19]. To our knowledge, since limited information is available on whether and how co-infections of intestinal parasites and malaria affect the specific immune response to malaria antigens [20], the goal of this study was to investigate the prevalence and relationship between co-infections of malaria (MAL+) and intestinal parasites (IP+) (infections with protozoans and/or helminths) on naturally acquired antibodies to malaria merozoite, as they are one of the main target sites for most vaccine candidates under clinical trials.

## Methods

### Ethical consideration

Ethical clearance used for the study was obtained from the Cameroon National Ethics Committee (IRB approval: N°2016/12/845/CE/CNERSH/SP). Administrative authorizations were obtained from authorities of the Ngali II and Mfou health districts (023/UYI/BTC/2016).

Participation in the study was voluntary with a written (English and French) informed consents obtained from parents of all participants after a clear explanation. A clinical examination was performed for all eligible participants by a medical doctor.

All participants positive for any *Plasmodium s*pp by RDT at the time of blood collection and those who were found to have PI by stool analysis were treated for free following the protocol recommended by the Cameroonian Ministry of Health by the medical doctor. All children with mild anaemia were given an iron supplement free of charge.

### Study area description

The study was conducted in Ngali II and Mfou, two villages in the central region of Cameroon (located at 4°27′N and 11°38′E) with a total population of about 1,000 children per squared Km (about 4000 in Ngali II and 6000 in Mfou) under the age of 15 years. The climate is typically equatorial with two discontinuous dry and rainy seasons. The annual average rainfall measures about 1600 mm$^3$ with an annual average temperature of 23°C [21].

Most children in Ngali II and Mfou over 3 years of age accompany their parents to the farm and return home late at night. The use of mosquito bed nets is rare in the two villages and residents have minimal access to portable water with approximately one well per 500 inhabitants. Currently, mass drug administration with albendazole is being performed twice a year by the Ministry of Health, that is usually conduced in schools and symptomatic cases are sent to the local clinic or hospital for follow up treatment.

### Study population

A cross sectional study was carried out in Ngali II and Mfou from January to May 2017, a transitional period from the dry to wet season. Children who had lived in either of the villages for at least six months and whose parents gave informed consent were included in the study. All participants were systematically examined by a physician for clinical systems of malaria and IP. Children who presented with symptoms of malaria, e.g., fever, headaches or intestinal

illnesses, e.g., diarrhea, vomiting were not enrolled. A total of 320 participants (140 from Ngali II and 180 from Mfou) aged 1–15 years participated in the study. Since both villages have the same demographic features, data for the two villages were combined.

## Blood collection and on-site testing for malaria

Venous peripheral blood (about 4mL) was collected by venipuncture using a butterfly needle (G22) and a 5mL labeled EDTA tube from all 320 participants. Haemoglobin (Hb) was measured using the HemoCue (AB Leo Diagnostics, Helsingborg, Sweden). On site, after collecting the venous blood from the participants, a drop from the same collected blood was placed on a CareStart™ Malaria pLDH/HRP-2 Combo Test (Access Bio Inc. USA) to detect histidine-rich protein-2 (HRP-2) specific to *Plasmodium falciparum* and Plasmodium lactate dehydrogenase (pLDH) pan-specific to *Plasmodium* spp. (P. *falciparum*, *P. vivax*, *P. malariae*, *P. ovale*). Results were read according to manufacturer instructions and recorded after 5 minutes.

## Laboratory detection of malaria parasites

Ten microliters of whole blood were used to prepare thick and thin smears for malaria parasite identification, speciation and quantification. The slides were air-dried overnight, and the thin blood smears were fixed in absolute (100%) methanol. Both thick and thin smears were stained using 10% Giemsa solution, washed with water and air-dried. Slides were then microscopically examined (thin and thick smear) for the presence of malaria parasites by two experienced microscopists. The parasite density was determined by counting the number of parasites against 200 leucocytes. The counts were expressed as the number of *P. falciparum*-infected erythrocytes (IE) per microliter of blood (Pf IE/μl), assuming an average leukocyte count of 8,000 cells/μl of blood [22]. When the difference in parasitaemia between the two readers was greater than 5 Pf IE/μl of blood, a third reader re-examined the slide and the mean of the two closest values were considered. Also, a differential count for eosinophil, lymphocytes, monocytes, neutrophils was obtained alongside parasitaemia and different malaria species (by microscopy)

## Antibody analysis

**The antigens (Ags).**   The recombinant Ags used included MSP1$_{42}$ of the FVO and 3D7 strains expressed in *Escherichia coli* and coupled at 0.2 μg per million beads, recombinant EBA-175 RII expressed in yeast coupled at 2.5 μg per million beads, recombinant MSP-3 C-terminal region expressed in *E. coli* coupled at 5 μg per million beads and recombinant MSP-2 (FC27 strain) coupled at 1 μg per million beads [23].

**The multiplex immunoassay.**   Plasma samples were tested for antibodies to the merozoite antigens using a multi-analyte platform assay with antigen-coupled to magnetic beads with different spectral addresses. Details of this assay used has been described previously [23] and optimized [24,25] In brief, plasma samples were diluted 1:100 with PBS, 50μl of plasma was incubated with 50μl antigen-coupled microspheres (2000 microspheres/test) for 60 minutes in the dark, washed with PBS, and then incubated at 500rpm for 60minutes at 25˚C on a rotating shaker and using a magnet plate separator. Then, 100 μl of secondary Ab (R-phycoerythrin-conjugated, Affini Pure F(ab′)$_2$ fragment, Goat anti-human IgG Fc fragment specific, Jackson Immuno-research, West Grove, PA, USA, Cat no. 109-116-170) diluted to 2 μg/mL in PBS-1% BSA was added to each well and incubated as above in the dark for 1 h. The mixture is then washed and a minimum of 100 beads were read in a MAGPIX® reader. A minimum bead count of 100 per spectral address recorded as Median Fluorescence Intensity (MFI).

Controls included on each plate were: PBS to determine background fluorescence, the negative control (NC) consisted of pooled plasma from four malaria-naïve US individuals, and the

positive control (PC) was pooled plasma from Cameroonians with high antibody levels to *Plasmodium falciparum*. Results were exported to Excel for analysis. The cut-off for positivity was calculated as mean of MFI + 3 standard deviation of the negative control as shown in the results sections.

## Stool sample collection and analysis

Sterile labelled stool collection vials were given to the parents along with instructions for proper stool collection. All samples were analyzed within 7 hours of collection to avoid missing hookworm eggs and minimize chances of under reporting. Approximately, 4 mg of feces was suspended in 5ml PBS and a drop examine by wet mount. The Kato Katz technique was used for morphological identification of helminths eggs, e.g., *A. lumbricoides*, *T. trichiura*, *or* larval stage of *Strongyloides stercoralis* [26] while the modified Ritchie's concentration stool technique was used to identify all protozoans and cestodes [27]. The smears were read at objective 10X for eggs and larvae and objective 40X for cysts and vegetative forms of protozoan. All stool slides were read by 2 technicians and in 2 different laboratories under supervision of a microbiologist and parasitologists.

## Data analysis

Data were analyzed using Microsoft Excel 2013, and GraphPad® prism 8. Standard summary statistics were used to describe the study population and results are presented as proportions. Fischer's exact test was used to compare antibody levels between the malaria-negative, IP-positive (MAL-,IP+) and malaria-positive, IP-negative (MAL+,IP-) groups, because of the small sample sizes of the groups. The one-way-ANOVA test was used to compare all 4 groups after checking for normality (e.g., age). An unpaired t test was used to compare the means of the MAL-,IP- vs. MAL+,IP- groups. Kruskal-Wallis test was used to compare antibody levels, which are not normally distributed, among the groups or within the MAL+IP+ groups. An individual was considered to have a co-infection if at least one IP species and *P. falciparum* were present. Anaemia was considered when Hb values were < 11.5 g/dL and classified according to WHO [28,29]. To search DNA sequences of *P. falciparum* EBA-175 and those of *E. histolytica* for possible cross-reactive epitopes, PfEBA175 (ncbi.nlm.nih.gov/gene/2654998) was compared with E. histolytica (ncbi.nlm.nih.gov/assembly/GCF_000208925.1) using Megablast for highly similar sequences and discontinuous megablast for more dissimilar sequences.

## Results

### The study population

A total of 320 children were enrolled (Table 1). Among the children, 76.3% were slide-positive for malaria (MAL+), with 59.4% having malaria without intestinal parasites (MAL+, IP-) and 16.9% being coinfected with malaria and intestinal parasites (MAL+, IP+). All subjects who tested positive for malaria using the rapid diagnostic field test were confirmed positive by microscopy. Among children who were infected with malaria, 71.3% were infected with only *P. falciparum* and 5% had *P. falciparum* and *P. malariae*. Interestingly, only 2.2% of the children had IP without malaria and 21.6% were negative for both malaria and IP.

The mean age of the children changed with infection status among the 4 groups (p = 0.0001) with the lowest age found in uninfected children (6.4 years) and highest in children with co-infections (9.3 years) (Table 1). Malaria infections were found in all age groups; whereas, none of the children under age 4 years had intestinal parasites. Mean haemoglobin levels were lower in children infected with malaria, but the difference was no significant

**Table 1. Description of 320 children infected with malaria and intestinal parasites (IP).**

|  | MAL-,IP- | MAL+IP- | MAL-,IP+ | Co-infections (MAL+,IP+) | P values |
|---|---|---|---|---|---|
| Number (%) of children | 69 (21.6) | 190 (59.4) | 7 (2.2) | 54 (16.9) | |
| Mean years of age (range) | 6.4 (1–14) | 7.9 (1–15) | 8.6(4–12) | 9.3(4–15) | 0.0001* |
| Parasitaemia: (median # infected erythrocytes/μl (range) | 0 | 420 (40–96,000) | 0 | 900 (40–30,970) | 0.1599** |
| Measures of anaemia | | | | | |
| Hb (g/dL) (mean ±SD) | 12.1 ±1.6 | 11.6 ± 2.2 | 12.2 ± 1.4 | 12.4±1.8 | 0.0658* |
| Prevalence of anaemia | | | | | |
| # (%) of children with Hb <11.5 g/dL | 21 (30.4) | 87 (45.8) | 2 (28.6) | 17 (31.5) | 0.0324*** |

*comparison among the 4 groups (ordinary one-way ANOVA).

** comparison among the 4 groups (Mann-Whitney test).

*** comparison between MAL-,IP- vs. MAL+,IP- (Fisher's exact test).

(p = 0.08; MAL-,IP- vs MAL+,IP-). The prevalence of anaemia was higher in children who were infected with malaria (MAL+,IP-)(p = 0.032), but not those with co-infections (p >0.999) compared to children who were parasite-negative (MAL-,IP-).

## Prevalence of intestinal parasites

Overall, 19.1% (61/320) of the children were positive for intestinal parasites, 16.9% of whom were also infected with malaria and 2.2% were IP+ but MAL- (Table 2). The most frequent helminthic parasites detected were *A. lumbrioides* (2.8%) and single cases of *T. trichiura* and *Strongyloides sp*. Among the 320 children, 14.7% had detectable protozoan infections, including 7.8% infected with *Giardia lamblia*, 5.9% with *E. histolytica complex*, and 0.9% with *Isospora sp*. No hookworm infections were detected in this study. Very few children had intestinal cestodes (Table 2). Interestingly, all of the children had single parasite infections, and polyparasitism was not found.

## Influence of age on malaria, intestinal parasites, anaemia and moderate eosinophilia

As expected, children aged 1 through 2 years did not have soil-transmitted IP and had normal eosinophil levels; whereas, 63% of 1-2-year old children were infected with malaria and had

**Table 2. Prevalence of intestinal parasites (IP+) in the 320 children, ages 1 to 15 years.**

|  | Number of Children | | |
|---|---|---|---|
|  | MAL-, IP+ | MAL+, IP+ | Total IP+(% positive) |
| Intestinal Parasites | | | |
| **Helminths** | | | **11 (3.4%)** |
| *Ascaris lumbricoides* | 2 | 7 | 9 (2.8%) |
| *Trichuris sp.* | 0 | 1 | 1 (0.44%) |
| *Strongyloides sp* | 0 | 1 | 1 (0.44%) |
| **Protozoans** | | | **48 (14.7%)** |
| *Giradia lamblia* | 3 | 22 | 25 (7.8%) |
| *Entamoeba histolytica complex* | 1 | 18 | 19 (5.9%) |
| *Isospora sp.* | 1 | 3 | 4 (0.9%) |
| **Cestodes** | | | **2 (0.63%)** |
| *Hymenolepis nana* | 0 | 2 | 2 (0.63%) |
| **Total IP** | **7 (2.2%)** | **54 (16.9%)** | **61** (19.1%) |

**Table 3. Influence of age on malaria, intestinal parasites, anaemia and percentage of peripheral eosinophils.**

| Age (years) | N = | % Mal+ | % IP+ | * % with anaemia | **% with eosinophilia |
|---|---|---|---|---|---|
| 1–2 | 27 | 63.0 | 0 | 55.6 | 0 |
| 3–4 | 47 | 61.7 | 6.4 | 48.9 | 4.3 |
| 5–6 | 40 | 62.5 | 15.0 | 40.0 | 7.5 |
| 7–8 | 63 | 88.9 | 28.6 | 36.5 | 9.5 |
| 9–10 | 55 | 83.6 | 21.8 | 38.2 | 20.0 |
| 11–12 | 54 | 79.6 | 25.9 | 38.9 | 22.2 |
| 13–15 | 34 | 82.4 | 23.5 | 26.5 | 38.3 |

*Anaemia: Children with haemoglobin less than 11.5 g/dL.

**Moderate eosinophilia: $\geq$1,500 eosinophils/mm$^3$ or $\geq$18.7% peripheral eosinophils.

the highest prevalence of anaemia (Table 3). In contrast, in children 9–15 years of age ~80% were slide-positive for malarial parasites, 24%-29% had intestinal parasites, and 10–38% had moderate eosinophilia. Thus, as children living in these villages increased with age, they began developing partial immunity to malaria symptoms and anaemia declined; whereas, the prevalence of IP and eosinophilia increased.

A comparison of anaemia and eosinophilia among the 4 groups of children shown in Table 1 was made (S1 Table). Results showed that anaemia was associated with malaria infections and eosinophilia was associated with IP.

## Antibody levels to malaria merozoite antigens

With repeated exposure to malaria, Ab prevalence and levels increased with age to the four merozoite antigens (Fig 1). Among 1- to 2-year-olds, only 25% of the infants had Ab to EBA-175 and MSP3, 30% had Ab to MSP2, but 80% had Ab to MSP1 (Fig 1). However, by age 13–15 years, 60% had acquired Ab to MSP3 and >80% had Ab EBA-175, MSP2 and MSP3 (Fig 1A). Among Ab-positive children, Ab levels also increased with age (Fig 1B–1E). Although different amounts of Ab were ultimately obtained to the different antigens, the overall trend was for an increase in median Ab with age. Thus, it was important to take age into consideration when making comparison between children infected with malaria (MAL+,IP-), co-infected with malaria and IP (MAL+,IP+) and those who were not infected (MAL-,IP-) at the time the study was conducted.

## Comparison of Ab levels in participants with and without malaria and IP

Since children below 3 years of age were not infected with IP, they were not included in the comparative studies described below. Given that Ab prevalence and levels increased with age, the study population was divided into 2 groups: children aged 3–10 years, a time period when children were becoming infected with IP (Table 3) and those 11–15 years, mainly children who had been infected repeatedly with malaria and may had lived with IP for a period of time. As predicted, Ab levels were slightly higher in MAL+ children due to current boosting compared to MAL-, but the differences were not significant (all p values >0.05) (Fig 2).

A comparison between Ab levels in children infected with malaria and co-infected with IP was conducted. Children with helminths and cestodes were not included in the analysis because the sample sizes were too small (n = 11). Regarding age, Ab levels were compared between children aged 3–10 years infected with malaria (n = 112) and co-infected with flagellate and intestinal protozoan (n = 25 children), including *G. lamblia* (n = 15) and *E. histolytica* (n = 10 children) (Fig 2). Antibody levels did not differ between malaria-infected children with

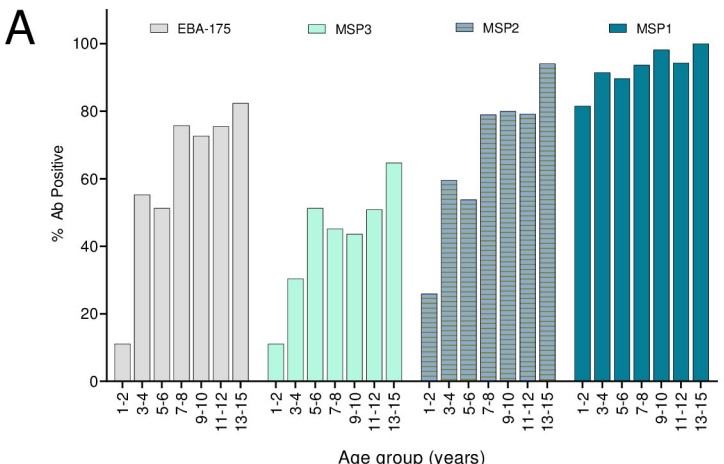

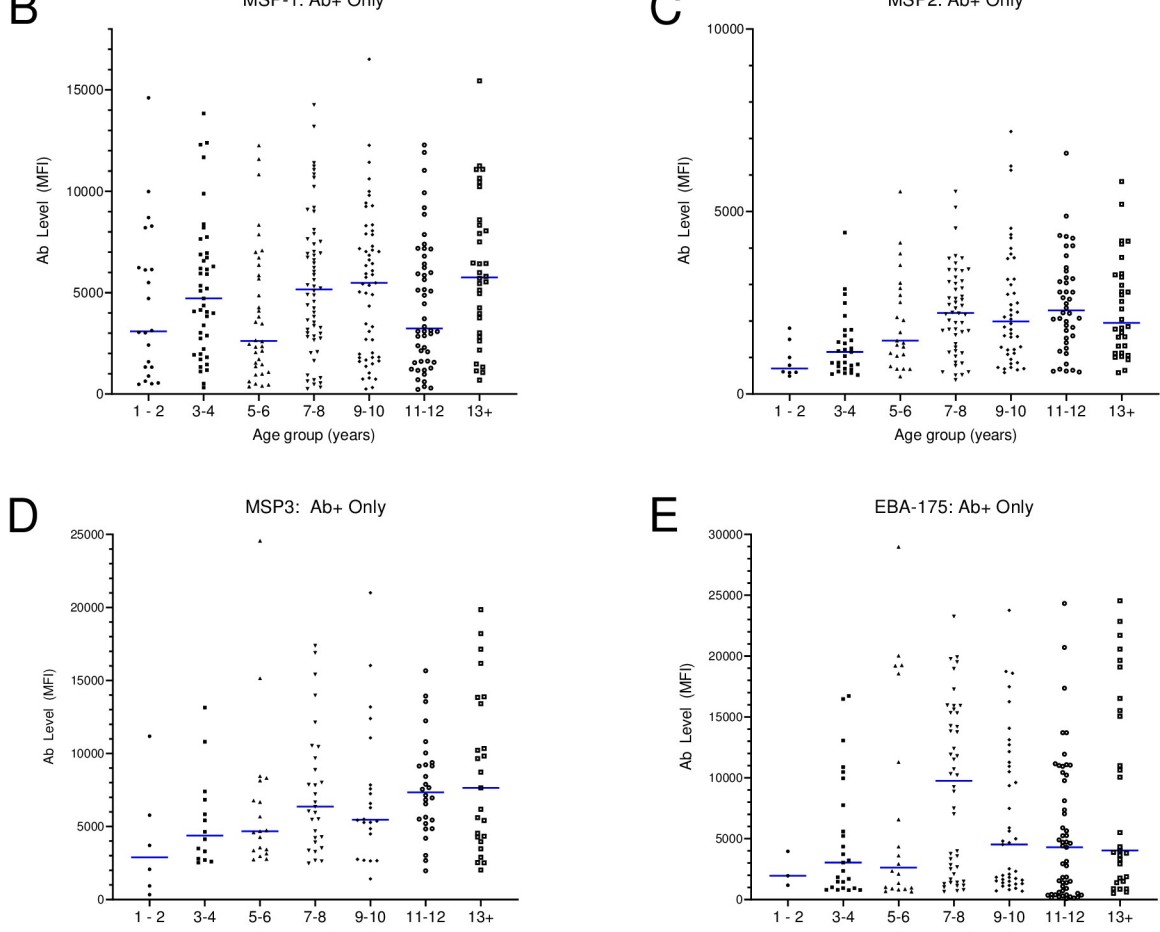

**Fig 1. Prevalence and amount of Ab in different age groups.** (A) Prevalence of Ab to the 4 merozoite antigens. The number of participants in each age group is provided in Table 3. Fig 1B–1E show Ab levels (MFI) for children who were Ab-positive for each age group. Horizontal bars represent median Ab levels. Kruskal-Wallis test (nonparametric comparison among groups) values were for MSP1 (p = 0.067); MSP2 (p<0.001); MSP3 (p = 0.086) and EBA (p = 0.056). MFI = Median fluorescence intensity; MSP = merozoite surface proteins; EBA = erythrocytes binding antigen.

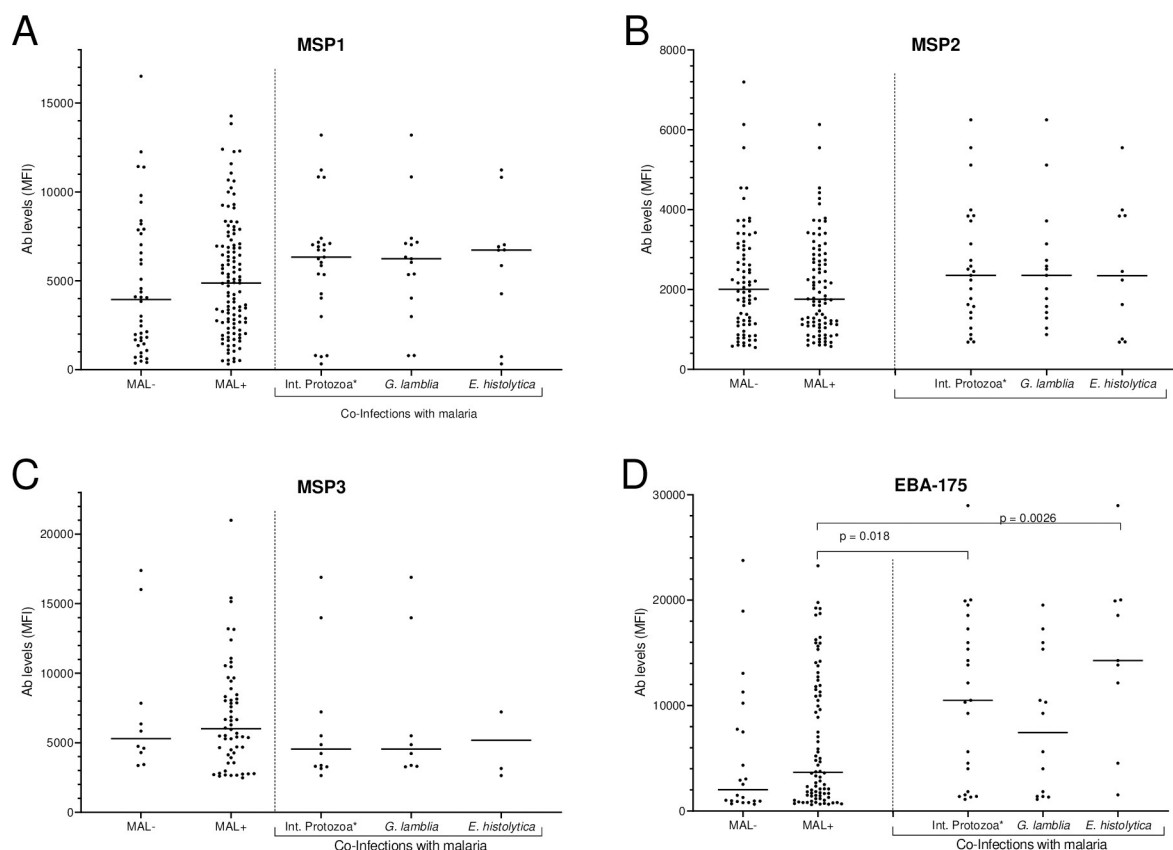

**Fig 2. Antibody levels in children ages 3 to 10 for all antibody-positive individuals.** Distribution of Ab levels in MFI among malaria negative (MAL-) and malaria-positive (MAL+) and those co-infected with malaria plus Intestinal (Int.) protozoa (n = 25); malaria plus *G. lamblia* (n = 15); and malaria plus *E. histolytica* (n = 10). The number of datapoints varied because not all participants had Ab to all antigens. Horizontal lines represent medians for the group. MFI = median florescence intensity; MSP = merozoite surface proteins, EBA = erythrocytes binding antigen (EBA).

or without intestinal protozoan for MSP1, MSP2 and MSP3; however, there were significantly higher Ab levels to EBA-175 in children co-infected with malaria and intestinal protozoans (p = 0.018) (Fig 2D). The higher Ab levels were due to *E. histolytica* infections (p = 0.0026), and not *G. lamblia* (p = 0.3844). No differences were found between children aged 11 to 15 years for any of the antigens between children with malaria (single infection) and co-infected with any of the IP.

To determine if higher Ab levels in children co-infected with *P. falciparum* and *E. histolytica* might be due to cross-reactive epitopes, a BLAST search for sequence homology between EBA-175 and *E. histolytica* proteins. No similarities were found using Metablast, and only one hit was found using discontinuous metablast which had a span of only 38 nucleotides (~13 amino acids) that had 82% similarity (Sequence ID: DS571535.1 Length: 11868. Range 1: 4999 to 5034. Score:35.6 bits (38), Expect:2.0, Identities:31/38(82%), Gaps:2/38(5%), Strand: Plus/ Minus.

Query 2053 `AAACAATACCAAGAATATCAAAAAGGAAATAATTACAA` 2090 Sbjct 5034 `AAAAAATA—AAGAAAATAAAAAAGGAAAAAATAACAA` 4999). Thus, there does not appear to be shared epitopes between these two pathogens that would explain the increase in Ab to EBA-175 in children with co-infections.

## Discussion

Malaria and polyparasitism (infections with protozoans and/or helminths) are still common conditions throughout Africa [30,31]. In the 1-15-year-old children living in the rural Cameroonian villages surveyed, the prevalence of slide-positive malaria was 76.3% and 19.1% for intestinal parasites, with 16.9% co-infections (Tables 1 and 2). This prevalence of malaria is similar to that found in other highly endemic regions of the country [32], and the prevalence of co-infections was 19.1%, which is similar to a prevalence of 18–27% reported in other regions of Cameroon [33,34]. This high transmission is related to geo-ecological and climatic conditions at the time of the study which was the transition from the dry to wet season, a period that favors vector breeding and distribution [35].

From Table 3, the prevalence of slide-positive malaria ranged from 61% to 90% in different age groups implying that children in these villages were repeatedly exposed to malaria throughout their lives. The prevalence of malaria in 2017 was similar to that recorded previously for Ngali II between 1998–2004, that ranged from 50% to 85% in 5–15 year old children with an estimated entomological inoculation rate of 0.7 infectious bites/per/night (~257 infectious bites annually) [36]. Prior studies have established that repeated exposure induces immunity to malaria, with development of anti-disease immunity followed by anti-parasite immunity [37–40]. As a result, the highest prevalence of 56% anaemia was found in young children [2,41,42] with a decline to 27% in 13 to 15-year-olds (Table 3). On the other hand, infections with IP only occurred later in life from 3 years. Increase in intestinal parasites was associated with an age-related increase in eosinophilia [43,44], a known innate immune response to helminthic and other soil-transmitted organisms (Table 3). In this study, only 11/320 (3.4%) children were infected with helminths. Although some epidemiological studies have demonstrated an increased risk of infection by *P. falciparum* in individuals co-infected with helminths, other results are conflicting [45,46]. The low prevalence of helminths in this study is explained, in part by, the fact that mass community de-worming is done biannually following the national infectious disease guide-line for IP control program. The most prevalent intestinal parasites were the protozoans, *G. lamblia* and *E. histolytica* [47–50]. These protozoa are commonly found in damp soil and contaminated water with a prevalence of 2–20% in Cameroon [51–54]. These results suggest children acquire their intestinal infections after learning to walk and interact with the environment. Thus, children in the study population were exposed to malaria early in life and began developing anti-malaria immunity prior to exposure to intestinal parasites.

Generally, both Ab prevalence and Ab levels increased with age in 1 to 15-year-olds living in this high transmission area (Fig 1). Often, the presence of Ab is used as markers of infection, including the merozoite antigens used in this study. The study compared antibody levels to *Plasmodium* merozoite antigens (MSP1, MSP2 MSP3, EBA17) between four groups of children, defined according to infection with malaria and/or intestinal parasites (MAL-IP-, MAL+IP-, MAL+IP+, MAL-IP+) [55–57]. Since over 80% of 1-2-year-olds had Ab to MSP1, humoral immunity began to develop early in life and continued to mature as children developed into adolescents (Fig 1). Often individuals who are MAL+ have higher Ab levels than MAL- individuals due to boosting of the Ab response [37,39,40]. In the current study, Ab levels did not differ significantly between MAL+ and MAL- individuals, neither those who were 3–10 years nor 11–15 years-old. This result was not surprising, since 75% of the children were slide-positive for malaria (Table 1). Because of high transmission, children are becoming infected almost on a daily basis and either are in the process of eliminating the new infection or reducing it to submicroscopic levels. Thus, most children living in areas with high perennial transmission will test positive for malaria by PCR. Because of constant re-exposure, the resulting immune response will be similar to that produced by a chronic infection.

This study measured the Ab response to 4 antigens present in *P. falciparum* merozoites, that play a role in immunity to malaria and are commonly used in studies of this type. Following schizont rupture, the merozoite surface antigens (MSPs) participate in initial attachment of merozoites to erythrocytes; then erythrocyte-binding antigen EBA-175 aids in binding and induces release of proteins localized in the micronemes that participates in junction formation [58]. Thus, the immune system of children living in Etoudi, Cameroon, would be exposed to these antigens simultaneously and repeatedly. Although considered to be vaccine candidates, Ab against these antigens are markers of past or current infection and not necessarily markers of protection. Antibodies against any one of the antigens alone is not an index of immunity to malaria. Thus, even if intestinal parasitic pathogens had down-regulated humoral responses to *P. falciparum*, reduced Ab levels would not necessarily mean the individual had increased susceptibility to malaria disease. Studying these antigens, however, provided a way to assess the impact of intestinal pathogens on humoral immunity to asexual *P. falciparum* parasites.

The results obtained in this study are similar to those reported previously in other areas of Cameroon. For example, Ab prevalence to merozoite antigens increase with age in high transmission areas in Cameroon like Etoudi; but the increase is at a different rate for different merozoite antigens [58,59] and varies in regions with different transmission rates. For example, in Buea and Limba, where EIR are >500 IB/P/year, 100% of children aged 5 years had Ab to AMA1, whereas, only 65% had Ab to MSP1 [60,61]. However, by age 15, 100% of children have Ab to 4 merozoite antigens studied. In addition to rate of malaria transmission, several factors influence the rate of Ab responses in children, e.g., the use of malaria intervention strategies. Cameroonian children less than 5years are generally more prone to anaemia and iron deficiencies, that may result in lower Ab levels to merozoite antigens [8]. Clearly, one major limitation of this study was that it was conducted in an area with MDA campaigns conducted bi-annually. Thus, the impact of IP helminthic infections could not be assessed. However, Ab levels and prevalence in Etoudi are similar to those in areas where MDA are given at different times. Thus, the current results reflect the situation in many regions in Cameroon.

Prior studies have demonstrated that malaria-helminths co-infections can down regulate malaria and orient the immune response via the Th2 response hence, making patients less sick [20,37,62,63]; whereas, others have demonstrated on the contrary that IP and malaria infections increase malaria disease [13,57]. Unfortunately, the current study could not resolve the controversy because very few children had helminthic infections, due to frequent treatment with albendazole via the mass drug administration (MDA) program conducted by the Ministry of Health and other random health campaigns. Despite the common practice of MDA in subtropical and tropical countries like Cameroon, complete eradication is usually not attained as other factors such as hygiene and sanitation, behavior of population and awareness are all require in an integrated manner to avert the problem, as these infections may return in as early as within four to six months post MDA [64,65]. Also, adverse events when not well managed may lead to rejection of treatment, thereby creating reservoirs for the parasites to continue to circulate in the community [66].

In our study population, co-infections with malaria and protozoans were relatively common. Ab levels to MSP1, MSP2 and MSP3 were similar in children infected with *P. falciparum* alone and those with protozoan (Fig 2); however, significantly higher Ab levels to EBA-175 were found in children co-infected with malaria and intestinal protozoan (p = 0.018). The higher Ab levels were due to *E. histolytica* infections (p = 0.0026), and not *G. lamblia* (p = 0.384). This result was unexpected. *E. histolytica* is a gut protozoan that cause both intestinal and extraintestinal infections such as amebic colitis (dysentery) and liver or brain abscess. In severe cases, this protozoan can cause a marked down-regulation of macrophage functions rendering the cells incapable of antigen presentation and unresponsive to cytokine stimulation

[67]. This decrease in macrophage function does not explain the increase in Ab to EBA-175. One possible explanation is that since malaria and *E. histolytica* are both protozoan pathogens, they might share common antigens, that is, EBA-175 could share homology with an *E. histolytica* antigen. To investigate this possibility, a blast search of the NCIB gene bank was conducted for EBA-175 and the *E. histolytica* genome. However, this search revealed only a ~13 amino acid sequence with 82% similarity, which seems unlikely to explain the increase in Ab levels of co-infected children. Finally, an alternative explanation could be that this result is a spurious observation by chance. Prior studies on Ab prevalence or levels to MSP1,2,3 in children infected with *E. histolytica* have not been reported. Clearly the association between malaria and *E. histolytica* merits further study.

Altogether, this observation needs to be repeated with a larger sample size as *E. histolytica* boosting of Ab to EBA-175 –co-infection might not only be limited to EBA-175, but other antigens as well. Children in these villages began to acquire an Ab response to the 4 merozoite antigens early in life, prior to infection with IP. There was no evidence that infection with IP influenced Ab levels or negatively-altered the already established Ab response to the 4 merozoite antigens.

## Conclusion

The prevalence of malaria was high in children 1–2 years old; whereas, intestinal parasite infections occurred in children over 3 years old. Thus, immunity to *P. falciparum* began prior to infection with soil-transmitted parasites. No differences were found in antibody prevalence or levels in malaria-infected and co-infected children, except antibody levels to EBA175 were significantly higher in children co-infected with malaria and *E. histolytica*. This is the first report of an interaction between malaria and *E. histolytica* and antibodies to EBA-175 and merits further evaluation.

## Supporting information

**S1 Table. Influence of co-infections on anemia and eosinophilia.**
(DOCX)

**S1 File.**
(XLS)

## Acknowledgments

We heartily thank all the participants and their parents who let them participate in the study in Ngali II and Mfou. We equally express our gratitude to the community health workers of these areas for their assistance during sample collection.

Anna Babakhanyan, University of Hawaii, provided the recombinant proteins (beads) used for the assays. The recombinant MSP antigens were contributed by the Malaria Vaccine Development Branch, NIAID, NIH and EBA-175 was provided by W. Bancroft through the PIBP, NIAID, NIH, Contracts Program.

The authors acknowledge the support of the entire staff of the Biotechnology Centre of University of Yaoundé I, Cameroon for their tremendous work in the field during sample collection and sample processing in the lab.

## Author Contributions

**Conceptualization:** Crespo'o Mbe-cho Ndiabamoh, Gabriel Loni Ekali.

**Formal analysis:** Crespo'o Mbe-cho Ndiabamoh, Livo Esemu.

**Funding acquisition:** Gabriel Loni Ekali, Rose Gana Fomban Leke.

**Investigation:** Crespo'o Mbe-cho Ndiabamoh, Jean Claude Djontu.

**Methodology:** Crespo'o Mbe-cho Ndiabamoh, Gabriel Loni Ekali, Livo Esemu.

**Project administration:** Crespo'o Mbe-cho Ndiabamoh.

**Resources:** Rose Gana Fomban Leke.

**Software:** Crespo'o Mbe-cho Ndiabamoh.

**Supervision:** Wilfred Mbacham, Jude Bigoga, Rose Gana Fomban Leke.

**Writing – original draft:** Crespo'o Mbe-cho Ndiabamoh.

**Writing – review & editing:** Crespo'o Mbe-cho Ndiabamoh, Yukie Michelle Lloyd, Diane
Wallace Taylor, Rose Gana Fomban Leke.

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
