## [Decision Letter · Decision Letter 0]

10 Jun 2020

PONE-D-20-12369

Full-title: Antibody Response to Malaria Merozoite Antigens in Asymptomatic Children Co-infected with Malaria and Intestinal Parasites

PLOS ONE

Dear Dr. Ndiabamoh,

Thank you for submitting your manuscript to PLoS ONE. After careful consideration, we felt that your manuscript requires substantial revision, following which it can possibly be reconsidered, thus governing the decision of a “major revision”. As requested by the reviewers, the authors need to address several concerns, particularly related to the study design, methods and results. A major concern raised by the Reviewers was because parasite detection was not followed by sensitive diagnostic techniques such as PCR-based assays, and submicroscopic infections may have been missed. Finally, the authors should follow the  policy of Plos One to share the raw data underlying their results.  Such policies help increase the reproducibility of the published literature, as well as make a larger body of data available for reuse and re-analysis. For your guidance, a copy of the reviewers' comments was included below.

We look forward to receiving your revised manuscript.

Kind regards,

Luzia Helena Carvalho, Ph.D.

Academic Editor

PLOS ONE

Journal Requirements:

"Funding Statement:

The beats used for the experiments were a gift from A. Babakhanyan (University of

Hawaii) and coupled by ELG. The MAGPIX used for the data analysis was MAGPX

I3038703, Luminex corporation 12212 technology Blvd Austin. Texas 78727. GFLR

provided the funds for the field work and analysis"

 "The author(s) received no specific funding for this work. Funding used in for this research was mentors (Prof Leke Rose) and a Gift of the magnetic beats from Dr Anna Babakhanyan. No other specific funding were received. "

Additionally, because some of your funding information pertains to [commercial funding//patents], we ask you to provide an updated Competing Interests statement, declaring all sources of commercial funding. 

In your Competing Interests statement, please confirm that your commercial funding does not alter your adherence to PLOS ONE Editorial policies and criteria by including the following statement: "This does not alter our adherence to PLOS ONE policies on sharing data and materials.” as detailed online in our guide for authors  http://journals.plos.org/plosone/s/competing-interests.  If this statement is not true and your adherence to PLOS policies on sharing data and materials is altered, please explain how.

* Please include the updated Competing Interests Statement and Funding Statement in your cover letter. We will change the online submission form on your behalf.

Reviewers' comments:

Reviewer's Responses to Questions

**Comments to the Author**

1. Is the manuscript technically sound, and do the data support the conclusions?

Reviewer #1: Yes

Reviewer #2: Yes

Reviewer #3: Partly

2. Has the statistical analysis been performed appropriately and rigorously? 

Reviewer #1: Yes

Reviewer #2: No

Reviewer #3: Yes

3. Have the authors made all data underlying the findings in their manuscript fully available?

Reviewer #1: Yes

Reviewer #2: No

Reviewer #3: Yes

4. Is the manuscript presented in an intelligible fashion and written in standard English?

Reviewer #1: Yes

Reviewer #2: Yes

Reviewer #3: Yes

5. Review Comments to the Author

Reviewer #1: Dr. Mbe-cho and colleagues sought to determine the prevalence of co-infection of malaria and intestinal parasites and its association with antibody levels to malaria merozoite antigens. The authors report that there was no difference in antibody prevalence or levels in malaria-infected and co-infected children, except antibody levels to EBA-175 were significantly higher in children co-infected with malaria and E. histolytica. Overall, the study is well-designed but these results do not significantly alter or impact our understanding of the association of malaria and helminths on antibody to malaria merozoite antigens.

1. The limitation of the study is that the parasite testing in children was not followed by sensitive diagnostic techniques like PCR, and light infections may have been missed which may have resulted in misclassification of the groups. Light infections may boost the antibody responses while children remain asymptomatic.

2. In this study, only 3.4% children were infected with helminths alone to get any meaningful data for antibody response to malaria in this group.

3. Very few children are positive for E. histolytica.

4. The data on the children's anthropomorphic measurements are not mentioned. Thus, there is not much point describing how they were collected.

5. There is no data on hookworm infection in the results.

6. The number of eggs per gram of stool were estimated for the parasites listed. Did the authors look at the responses in children with high or low intensity of the parasites?

7. Table 2 is not necessary, it can be written as text.

8. Page 21, reference # 54, year of publication is missing.

Please check spelling and typographical errors scattered through the manuscript (page and lines are given from word document):

1. Page 2, line 3, change led to lead in the sentence.

2. Page 2, line 14, correct the spelling of Rietchi concentration method

3. Page 6, line 21: The bracket has to be closed here: (AB Leo Diagnostics, Helsingborg, Sweden.

4. Page 7, line 17 and 18: Correct 50ul to 50µl

5. Page 9 and 10: In the text, the p value for anemia (MAL+,IP-) is p=0.034; p value for the same in Table 1 is p=0.032; it needs to be corrected.

6. Page 10: In Table 1, % sign is missing in column 5 for children with Hb.

7. Page 10, line 3: In the sentence, change major to majority.

8. Page 14, line 27: In the sentence, MSL- should be MAL-

9. Page 17, line 15: change beats to beads

Re-write the following sentences, they are not very clear:

Page 4, line 8:

However, with most children getting infected with several episodes of infections in a short period, this renders them more prone to having clinical symptoms since the immune systems doesn’t fully recover.

Page 4, line 20:

Concomitant infections in humans have suggested that Ascaris lumbricoides infection may protect against cerebral malaria (11,12), while other studies, children infected by S. mansoni were more susceptible to P. falciparum infection and develop acute malaria episodes.

Page 15, line 3:

In essence, the immune response in individuals who are repeatedly infection would be similar to that produce during chronic infections.

Reviewer #2: The answer to the questions is divided into Major comments, Minor comments. Additionally, I wrote minor observations that, I hope, will help this manuscript to improve readability and consistency.

1. Is the manuscript technically sound, and do the data support the conclusions?

2. Has the statistical analysis been performed appropriately and rigorously?

3. Have the authors made all data underlying the findings in their manuscript fully available?

4. Is the manuscript presented in an intelligible fashion and written in standard English?

Major comments:

• Given that there were no differences in the IgG response between age groups, it would be interesting to join these data, evaluate all the coinfected individuals, and then split the data into Giardia, E. hystolitica.

• I strongly suggest dividing the age of individuals in 0-5, 5-10, 10-15 years-old to partially solve the "N" problem of the groups.

• Because of the absence of molecular Diagnosis and considering that the authors mention the possibility of oh having low parasitemia infections in the MAL- group. It is important to include MAL- individuals in Figure 1.

• It is necessary to compare parasite data with similar regions in Cameroon. Please compare and cite:

• (Malaria and Helminth Co-Infection in Children Living in a Malaria Endemic Setting of Mount Cameroon and Predictors of Anemia from Theresa K Nkuo-Akenji et al. 2006)

• Malaria, Helminths, Coinfection and Anaemia in a Cohort of Children From Mutengene, South Western Cameroon from Clarisse Njua-Yafi et al. 2016.

• Do the authors have information about malaria and intestinal parasites last treatments? On page 17, it was commented that Albendazole treatment was frequent in these children. Deworming information will help the readers to understand why the prevalence of intestinal parasites was low compared with other studies in Cameroon. Additionally, reinforce in the discussion section that collecting/reporting that information is valuable for coinfection studies.

• (Figure 1 B, C, D, E) use the same scale limits for all plots. This is also useful to understand differences in levels of antigenicity between proteins.

• (table 3) How could the authors explain increased eosinophilia with low levels of helminth infection? This mainly applies to the age group > 9 years-old.

• (Page 17) The authors argue, "First, children living in moist or wet environments where mosquitoes breed and E. histolytica are more abundant would have a high risk of acquiring both infections, that would result in frequent boosting of the Ab response." This explanation for intestinal parasite influence on antibody production alteration is not viable since Giardia's frequency is higher than E. hystolitica in the studied population.

• (Page 17) The affirmation "Secondly, since malaria and E. histolytica are both amoebae, they might share common antigens, for example, EBA-175 could share homology with an E. histolytica antigen." is false. Plasmodium falciparum is not an is a protozoan. This group belongs to Apicomplexa organisms. For that reason, the hypothesis about correlating Plasmodium and E. hystolitica is wrong.

• How different are the two Villages Ngali II and Mfou in the central region of Cameroon? Does it exist a difference in humidity and soil moist, once the authors claimed that this variable could explain differences of Entamoeba histolytica?

Minor comments:

• What criteria were used to divide the population into seven groups according to age?

• Please specify how anthropometric parameters were used in the study, once they were described but not used in the study. If this information was not used, please remove these sentences.

• Has the studied region presence of Schistosoma haematobium? If the authors have register if this parasite in the area, Did they examined urine samples to discard infections with this parasite?

• Were the individuals asymptomatic to intestinal parasites infection too? No diarrhea, abdominal pain, etc.? Please clarify.

• (Page 6) It was mentioned that Plasmodium parasitemia was quantified. Did the authors observe any correlation between the Plasmodium parasite burden and the levels of IgG responses to the antigens?

• (End of Page 7) Please specify: If the cut-off is MFI+3*SD, how the standard deviation was calculated if the negative controls were pooled? Was this experiment repeated or used replicates? Traditionally, the negative controls are tested simultaneously in different wells of the plate, and the cut-off is calculated from those values.

• Did the authors analyze the effect of helminth parasite burden (number of eggs/gram of stool) in those individuals with helminths? This valuable information was commented on but never included in the analysis. If not used,e I do not see the necessity of describing in the methods section

• For data analysis:

• Before using ANOVA, did the authors checked for the normality of the variables? If yes, please specify, if not, calculate the normality of the variables and the other ANOVA assumptions.

• If the authors have not-normal variables, they should use the Kruskal-Wallis non-parametric, and Dunn posthoc tests to verify differences between groups.

• Please check frequencies described in table 1 (MAL+IP- 58.8%) vs. the values reported in the second line page 9. (59.4%).

• Sum of 58.8%+16.9% = 75.7% not 75.6%.

• In table 1, please add a column with P-values to facilitate the interpretation of the differences between groups. Please report statistics of multiple comparisons between groups too.

• What is the potential hypothesis to explain the increased values of parasitemia in the coinfected group?

• Please comment in the text the presence of multiparasitism in the studied individuals.

• (Page 11 table 3). Please include values of anemia and eosinophilia in individuals coinfected. In the current configuration is constructed is hard to determine the coinfection impact in anemia and eosinophilia values.

• (Page 11). In the sentence, "Thus, as children living in these villages increased with age, they developed partial immunity to malaria and anemia declined; whereas, the prevalence of IP and eosinophilia increased." In this sentence, it is necessary to specify that "protection" is protection against malaria symptoms. The table clearly shows that the frequency of malaria does not decrease with age, only the anemia.

• Please plot Age vs. Antibody levels for each protein to verify the correlation for each protein studied.

• As an exploratory analysis, I suggest joining all data and make a boxplot comparing MFI between MAl-PI-, MAL-PI+, MAL+PI-, and MAL+PI+. Mainly for MSP1, MPS2, and MSP3 group age 3-10 and 11-15 to check.

• The sentence "E. histolytica is a gut amoeba that causes both intestinal and extraintestinal infections such as amebic colitis (dysentery) and liver or brain abscess. The protozoa cause a marked down-regulation of macrophage functions rendering the cells incapable of antigen presentation and unresponsive to cytokine stimulation (57)" does not explain the increase of antibody production in E. histolytica infected group. Why could a diminishing antigen presentation generate higher levels of anti-Plasmodium antigens?

Other observations/questions:

• In the title, add "IgG" to Antibody response.

• Check all scientific names of parasite species for correct formatting in italics. (Example Entamoeba histolytica in the Results section in the abstract)

• Please, mention in the background the region where the study was performed.

• It is necessary to describe and discuss the role of MSP1, MPS2, MSP3, and EBA-175 as markers in serological studies.

• Considering that coinfection prevalence is relatively low, I consider that it is important to discriminate with colors or point shapes the individuals MAL-IP-, MAL+IP-, MAL-IP+, MAL+IP+ in Figure 1 B-C-D-E

• In page 6 subtitle "Laboratory detection, quantification and speciation of malaria parasites.", I will not use speciation here. I suggest "Diagnosis and quantification of Plasmodium sp. parasites.

• (Page 14-15) What type of parasite is "Amoeba"? What is the difference between "Amoeba" and E. hystolitica? Traditionally, E. hystolitica is considered an amoeba too.

• In table 1, to facilitate reading, please remove symbols % and /ul located in cells with data and add to the columns describing the variables.

• For consistency, unify parasitemia vs. parasitaemia, anemia vs. anaemia in the text and plots.

• (Page 10) change "The major of helminth parasites" to "The most frequent helminth species detected."

• (table 2) Check all the total numbers for the "Total IP+" column. For example, for protozoans, the sum is 29+19+4 = 48, and it was reported 47

• (Page 13) In plot titles Change Ab (Antibody) to IgG

• (Figure 1E) Add, Change from EBA to EBA-175.

• Please verify all references formatting (For example, reference 42 is all in capital letters)

Reviewer #3: Review Comments to the Author

Please find attached the manuscript with my comments for the manuscript 'Full-title: Antibody Response to Malaria Merozoite Antigens in Asymptomatic Children Co-infected with Malaria and Intestinal Parasites'

6. PLOS authors have the option to publish the peer review history of their article (what does this mean?). If published, this will include your full peer review and any attached files.

Reviewer #1: No

Reviewer #2: No

Reviewer #3: No

---

## [Author Response · Author response to Decision Letter 0]

23 Jul 2020

Reviewer #1: Dr. Mbe-cho and colleagues sought to determine the prevalence of co-infection of malaria and intestinal parasites and its association with antibody levels to malaria merozoite antigens. The authors report that there was no difference in antibody prevalence or levels in malaria-infected and co-infected children, except antibody levels to EBA-175 were significantly higher in children co-infected with malaria and E. histolytica. Overall, the study is well-designed but these results do not significantly alter or impact our understanding of the association of malaria and helminths on antibody to malaria merozoite antigens.

1. The limitation of the study is that the parasite testing in children was not followed by sensitive diagnostic techniques like PCR, and light infections may have been missed which may have resulted in misclassification of the groups. Light infections may boost the antibody responses while children remain asymptomatic.

Reply: We understand the concern. When the study was conducted (2017) in the rural villages, the prevalence of slide-positive malaria was 75.6%. In a prior study conducted in the village (Leke et al 2010), an equivalent prevalence was found of P. falciparum (50-85%) in children aged 5-15 years over a 5-year period. The estimated entomological inoculation rate (EIR) was 0.7 infectious bites/person/ nightly thought out the year (~257 IB/P/Y). Based on the more recent malaria prevalence, it appears that the current EIR is similar. Thus, children were most likely being bitten approximately every-other night by an infectious mosquito, since bednets were not routinely used. With this high level of transmission, most of the slide-negative children would be PCR-positive for malaria, i.e., have enough immunity to reduce malaria to submicroscopic levels. Unfortunately, in very high transmission areas like the one reported herein, everyone will have some circulating P. falciparum parasites. So, classifying subjects as slide-positive vs slide-negative may not reflect presence/absence of parasites, but provide information on the immune status of the person. In revising the MS, information from the study by Leke et al. was included as well as a discussion of submicroscopic infections in the revised Discussion.

2. In this study, only 3.4% children were infected with helminths alone to get any meaningful data for antibody response to malaria in this group.

Reply: We agree, the sample size of children with helminth infections is too small to provide meaningful information. Accordingly, Ab levels in children with helminth infections were not analyzed. To explain the low prevalence of helminths, information on the Ministry of Heatlth’s policy for biannual treatment of children for worms was provided. 

3. Very few children are positive for E. histolytica.

Reply: True, the prevalence of Entamoeba in our study was only 5.9%, which is lower than that reported in studies in these areas of ~23% (T. E. Kwenti et al., 2016). In our study, the prevalence was lower, probably due to rigorous mass drug administration (MDA) programs implemented by the Ministry of Health and other regular or seasonal health campaigns. 

4. The data on the children's anthropomorphic measurements are not mentioned. Thus, there is not much point describing how they were collected.

Reply: This section was removed from the Methods section. 

5. There is no data on hookworm infection in the results.

Reply: The prevalence of hookworm infections was considered in this study during stool exams and, surprisingly, we did not find hookworms in the samples collected, most likely due to regular deworming and improved hygiene in the area. No invasive methods were used for diagnosis of adult worms. From a paper published by E. Kwenti et al. (2016) the prevalence of hookworm was 7% in south west region Cameroon. 

6. The number of eggs per gram of stool were estimated for the parasites listed. Did the authors look at the responses in children with high or low intensity of the parasites?

Reply: In this study, after obtaining the prevalence of parasites and comparing with antibody response, no significant difference was observed between the malaria antibodies levels and parasites eggs counts. 

7. Table 2 is not necessary, it can be written as text.

Reply: Thanks for the comment, but we think Table 2 summarizes the data more clearly and allows readers to easily compare results from different groups than presenting them in the text. Table 2 has been revised.

8. Page 21, reference # 54, year of publication is missing.

Reply: Year of publication has been included.

Please check spelling and typographical errors scattered through the manuscript (page and lines are given from word document):

1. Page 2, line 3, change led to lead in the sentence.

Reply: The word “led” has been changed to “lead”. 

2. Page 2, line 14, correct the spelling of Rietchi concentration method

Reply: Spelling has been corrected to “Ritchie” 

3. Page 6, line 21: The bracket has to be closed here: (AB Leo Diagnostics, Helsingborg, Sweden. 

Reply: The bracket has been closed. 

4. Page 7, line 17 and 18: Correct 50ul to 50µl

Reply: The change has been made. 

5. Page 9 and 10: In the text, the p value for anemia (MAL+,IP-) is p=0.034; p value for the same in Table 1 is p=0.032; it needs to be corrected.

Reply: P value has been corrected to P=0.032 (correct value) in the text. 

6. Page 10: In Table 1, % sign is missing in column 5 for children with Hb.

Reply: The % symbol has been included in table 1, column 5.

7. Page 10, line 3: In the sentence, change major to majority.

Reply: The word “major” has been changed to “majority”. 

8. Page 14, line 27: In the sentence, MSL- should be MAL-

Reply: In Line 27 of page 14, MSL- has been changed to MAL-

9. Page 17, line 15: change beats to beads

Reply: The spelling of beads has been corrected.

10. Re-write the following sentences, they are not very clear:

Page 4, line 8:

However, with most children getting infected with several episodes of infections in a short period, this renders them more prone to having clinical symptoms since the immune systems doesn’t fully recover.

Reply: The sentence has been deleted because the information is not directly relevant to the study.

Page 4, line 20:

Concomitant infections in humans have suggested that Ascaris lumbricoides infection may protect against cerebral malaria (11,12), while other studies, children infected by S. mansoni were more susceptible to P. falciparum infection and develop acute malaria episodes.

Reply: The sentence has been revised to read: “Studies on concomitant infections in humans suggest that A. lumbricoides infection may protect against cerebral malaria (11,12), while other studies suggest that children infected by S. mansoni may be more susceptible to P. falciparum infections and develop acute malaria episodes (13,14).”

Page 15, line 3:

In essence, the immune response in individuals who are repeatedly infection would be similar to that produce during chronic infections.

Reply: To clarify the statement, the text has been revised to read: “Because of high transmission, the children are becoming infected almost daily and are either in the process of eliminating the new infection or reducing it to a submicroscopic level. Because of constant re-exposure, the resulting immune response will be similar to that produced by a chronic infection.

Reviewer #2: The answer to the questions is divided into Major comments, Minor comments. Additionally, I wrote minor observations that, I hope, will help this manuscript to improve readability and consistency.

1. Is the manuscript technically sound, and do the data support the conclusions?

2. Has the statistical analysis been performed appropriately and rigorously?

3. Have the authors made all data underlying the findings in their manuscript fully available?

4. Is the manuscript presented in an intelligible fashion and written in standard English?

Major comments:

• Given that there were no differences in the IgG response between age groups, it would be interesting to join these data, evaluate all the coinfected individuals, and then split the data into Giardia, E. hystolitica.

Reply: We are confused by this comment, because Fig 1 shows an increase in both Ab prevalence (Fig. 1A) and Ab levels (Fig 1 B-E) with age in Ab-positive children (Kruskal-Wallis test p values were p<0.001 MSP2 and p=0.05-0.086 (borderline) for the other antigens). 

 We believe combining all MAL+,IP+ children into single a group is unwise, since they were infected with a conglomerate of intestinal helminths, cestodes and protozoa (see Table 2). Combining children with such heterogenous infections is unlikely to provide meaningful information. 

• I strongly suggest dividing the age of individuals in 0-5, 5-10, 10-15 years-old to partially solve the "N" problem of the groups.

Reply: Thanks for the comment. Initially, children were groups into 5-year categories as suggested by the Reviewer, i.e., 0-5, 5-10, 10-15 years old. However, when the data set showed that children aged 1 to 2 did not have intestinal parasites, the results were grouped into 2-year intervals, that allowed us to more closely define the increase in Ab prevalence (Fig. 1A) and Ab levels (Fig 1 -B,C,D,E) with age. The purpose of Fig 1 was to determine if age was a variable that needed to be taken into consideration during data analysis.

• Because of the absence of molecular Diagnosis and considering that the authors mention the possibility of oh having low parasitemia infections in the MAL- group. It is important to include MAL- individuals in Figure 1.

Reply: We are sorry if we didn’t make the point clear. ALL children who were Ab-positive are included in Fig 1, including those who are MAL+ and MAL-. Because malaria transmission is high in the area, all children in the study had been exposed to P. falciparum and many of the MAL- children were Ab-positive. 

• It is necessary to compare parasite data with similar regions in Cameroon. Please compare and cite:

• (Malaria and Helminth Co-Infection in Children Living in a Malaria Endemic Setting of Mount Cameroon and Predictors of Anemia from Theresa K Nkuo-Akenji et al. 2006)

• Malaria, Helminths, Coinfection and Anaemia in a Cohort of Children From Mutengene, South Western Cameroon from Clarisse Njua-Yafi et al. 2016.

Reply: We thank the Reviewer for pointing out the omission of key references. Information from these studies have been included in the revised Discussion. The text now reads, “…..to those found in other highly [malaria] endemic regions of the country (32), and the prevalence of co-infections was 19.1%, which is similar to the prevalence of co-infections of 18 – 27% reported in other regions of Cameroon (9,44). The references have been added to the reference section. 

• Do the authors have information about malaria and intestinal parasites last treatments? On page 17, it was commented that Albendazole treatment was frequent in these children. Deworming information will help the readers to understand why the prevalence of intestinal parasites was low compared with other studies in Cameroon. Additionally, reinforce in the discussion section that collecting/reporting that information is valuable for coinfection studies.

Reply: In response to the Reviewer’s suggestion, the following information has been added to the Methods section. “Currently, mass drug administration with albendazole is being performed twice a year by the Ministry of Health, that is usually conduced in schools and symptomatic cases are sent to the local clinic or hospital for follow up treatment.”

• (Figure 1 B, C, D, E) use the same scale limits for all plots. This is also useful to understand differences in levels of antigenicity between proteins.

Reply: We understand the comment, but we do not wish to change the Y-axis on Fig 1, since it is risky to make a direct comparison of Ab levels between antigens in serological assays. A number of variables, including parasite strain, the system to produce recombinant proteins, protein purity, the amount of antigen used, number of exposed epitopes, dilution of plasma, etc., influence the overall results. Even when Luminex beads are covalently-coupled with saturating amounts of antigen, it is questionable if direct comparison of MFI can be made between antigens. Although our assays have been optimized and equivalence amounts of antigen used during bead-coupling, comparisons among the antigens may not provide accurate information about immunogenicity. In Figs1 B, C, D, E, the Y-Axis was selected to show the best distribution of the MFI results. 

• (table 3) How could the authors explain increased eosinophilia with low levels of helminth infection? This mainly applies to the age group > 9 years-old.

Reply: After age 2, children start becoming infected with helminths, resulting in an increase in eosinophil counts. During the biannual drug treatment campaign, helminthic infections are eliminated, but eosinophilia persists for a period of time. With increasing age, more children in the area become i) infected and ii) re-infected, resulting in an increase in prevalence of eosinophilia. 

• (Page 17) The authors argue, "First, children living in moist or wet environments where mosquitoes breed and E. histolytica are more abundant would have a high risk of acquiring both infections, that would result in frequent boosting of the Ab response." This explanation for intestinal parasite influence on antibody production alteration is not viable since Giardia's frequency is higher than E. histolytica in the studied population.

Reply: The sentence has been deleted from the Discussion. 

• (Page 17) The affirmation "Secondly, since malaria and E. histolytica are both amoebae, they might share common antigens, for example, EBA-175 could share homology with an E. histolytica antigen." is false. Plasmodium falciparum is not an is a protozoan. This group belongs to Apicomplexa organisms. For that reason, the hypothesis about correlating Plasmodium and E. histolytic is wrong.

Sorry, “amoebae” was a typo. Both Plasmodium falciparum and E. histolytica are protozoans. The Discussion has been revised to read “parasitic protozoa.” 

• How different are the two Villages Ngali II and Mfou in the central region of Cameroon? Does it exist a difference in humidity and soil moist, once the authors claimed that this variable could explain differences of Entamoeba histolytica?

Reply: The two villages are very similar with no major differences in humidity or soil moisture. The estimated annual average rainfall measures 1600 mm3 with an annual average temperature of 23°C for Ngali II and for Mfou. According to the National Meteorology agency, the average humidity for the center regions is 83%. Ngali and Mfou are both in the center region of Cameroon about 60km apart. Note: as mentioned above, the words “humidity and soil moisture” have been deleted from the MS.

Minor comments:

• What criteria were used to divide the population into seven groups according to age?

Reply: The fact that Intestinal parasite (IP) infections was only observed in children >2 years, helped guide separation of the children into seven groups.

• Please specify how anthropometric parameters were used in the study, once they were described but not used in the study. If this information was not used, please remove these sentences.

Reply: The sentence has been removed. 

• Has the studied region presence of Schistosoma haematobium? If the authors have register if this parasite in the area, Did they examined urine samples to discard infections with this parasite?

Reply: Detection of S. haematobium was not included in the study design because of low prevalence in the study area. A study conducted in this area (and other regions of Cameroon) by Louis-Albert Tchuem Tchuenté et al., (2012) reported a prevalence of S. haematobium of only 1.72%. Since a large sample size would be required to assess the impact of this pathogen on the Ab response to malaria, S. haematobium was not included in the study. 

• Were the individuals asymptomatic to intestinal parasites infection too? No diarrhea, abdominal pain, etc.? Please clarify.

Reply: Yes. To make the point clear, the Methods section has been revised and states that all children with clinical cases of malaria or intestinal parasites were not included in the study and referred to the local clinic/hospital by the attending physician for treatment. Thank you for the comment.

• (Page 6) It was mentioned that Plasmodium parasitemia was quantified. Did the authors observe any correlation between the Plasmodium parasite burden and the levels of IgG responses to the antigens?

Reply: As expected, there was no correlation between parasitemia and malaria antibody levels. 

• (End of Page 7) Please specify: If the cut-off is MFI+3*SD, how the standard deviation was calculated if the negative controls were pooled? Was this experiment repeated or used replicates? Traditionally, the negative controls are tested simultaneously in different wells of the plate, and the cut-off is calculated from those values.

Reply: Pooled negative control plasma sample were run in triplicates on the same plates as the test samples in all experiments, as well as the positive controls. The cut-off was obtained by calculating MFI+3 SD of the triplicates on all plates in the experiment. 

• Did the authors analyze the effect of helminth parasite burden (number of eggs/gram of stool) in those individuals with helminths? This valuable information was commented on but never included in the analysis. If not used,e I do not see the necessity of describing in the methods section

Reply: The information has been deleted from the Methods section. 

• For data analysis:

• Before using ANOVA, did the authors checked for the normality of the variables? If yes, please specify, if not, calculate the normality of the variables and the other ANOVA assumptions.

Reply: Yes, ANOVA was used to compare difference in age across the 4 groups (Table 2). However, comparisons of Ab MFI, which are not normally distributed, with age (Fig. 1) were performed using the Kruskal-Wallis test. The Methods section (Data analysis) has been revised. Information in Fig. 1 legend was correct.

• If the authors have not-normal variables, they should use the Kruskal-Wallis non-parametric, and Dunn posthoc tests to verify differences between groups.

Reply: Sorry for the mistake in the Methods section. The Kruskal-Wallis nonparametric test was performed in Fig 1 and 2. A posthoc test was not performed, as the goal was not to determine when peak Ab levels were obtained, but to determine if age had an influence on Ab levels. Since age was a variable, data for all age groups could not be combined, but rather age was taken into consideration during data analysis.

• Please check frequencies described in table 1 (MAL+IP- 58.8%) vs. the values reported in the second line page 9. (59.4%). 

Reply: 59.4% is the correct value. The text has been revised.

• Sum of 58.8%+16.9% = 75.7% not 75.6%. 

 Reply: Thank you for catching the error. The values in Table 1 and text have been revised and are now consistent. 

• In table 1, please add a column with P-values to facilitate the interpretation of the differences between groups. Please report statistics of multiple comparisons between groups too.

Reply: The comparisons requested by the reviewer were originally provided in the Table legend. To comply with the request, the p values have been moved to a column labeled “p values” and the method of analysis was retained in the Table legend. 

• What is the potential hypothesis to explain the increased values of parasitemia in the coinfected group?

Reply: There is no significant difference in parasitemia between the two groups (p=0.1599). In fact, the higher parasitemia was found in young children who were intestinal parasite-negative (probably because very young children were in this group).

• Please comment in the text the presence of multi-parasitism in the studied individuals.

Reply: We thank the reviewer for the comment. The following sentence has been added to the Results section. “Interestingly, all of the children had single parasite infections, and polyparasitism was not found.”

• (Page 11 table 3). Please include values of anemia and eosinophilia in individuals coinfected. In the current configuration is constructed is hard to determine the coinfection impact in anemia and eosinophilia values.

Reply: Table 3 was designed to evaluate the influence of age on malaria, IP, anemia and eosinophilia. The number of co-infections are too small to be divided by age. In an attempt to address the Reviewer’s comment, a separate Table was designed that compares the influence of no infections, malaria-positive only, and co-infections on percent with anemia and eosinophilia. The Table will be up-loaded as supplemental Table 1. It essentially showed that same results as expected, anemia was associated with malaria and eosinophils were associated with co-infections.

• (Page 11). In the sentence, "Thus, as children living in these villages increased with age, they developed partial immunity to malaria and anemia declined; whereas, the prevalence of IP and eosinophilia increased." In this sentence, it is necessary to specify that "protection" is protection against malaria symptoms. The table clearly shows that the frequency of malaria does not decrease with age, only the anemia.

Reply: The sentence has been revised to read: “Thus, as children living in these villages increased with age, they began developing partial immunity to malaria symptoms and anemia declined; whereas, the prevalence of IP and eosinophilia increased.

• Please plot Age vs. Antibody levels for each protein to verify the correlation for each protein studied. 

Reply: The figure on the right confirms that Ab levels increase with age. The figure shows a linear regression analysis of Ab levels for MSP1, MSP2, MSP3 and EBA-175 using data from all 320 children, and includes the equation for the regression line, the R2 value (all positive), and p value (all significant). Thus, the figure confirms that Ab levels increase with age. We do NOT wish to include this figure in the MS since it is essentially identical to the one shown in Fig 1 B, C, D and E. In fact, we feel that the information in Fig 1B-E is easier for the reader to understand. 

Note: If the figure is not shown, it is provided in a separate document. 

• As an exploratory analysis, I suggest joining all data and make a boxplot comparing MFI between MAl-PI-, MAL-PI+, MAL+PI-, and MAL+PI+. Mainly for MSP1, MPS2, and MSP3 group age 3-10 and 11-15 to check.

Reply: We thank the Reviewer Thanks for the suggestion concerning exploratory analysis. A comparison of Ab levels in two of the above groups (MAL-,IP-, and MAL+,IP-) is shown in Fig 2. Unfortunately, the number of children in the MAL-,PI+ group is too small to provide valuable information. As stated above, children in the MAL-,PI+ group (n=54) are infected with a variety of intestinal helminths, cestodes and protozoa (see Table 2). With such a diverse range of pathogens, plotting the data as a boxplot will not provide useful information. In Fig. 2, the distribution of Ab levels in children co-infected with malaria and single intestinal pathogens is provided. We feel this approach is more informative than “dumping all pathogens together.” 

• The sentence "E. histolytica is a gut amoeba that causes both intestinal and extraintestinal infections such as amebic colitis (dysentery) and liver or brain abscess. The protozoa cause a marked down-regulation of macrophage functions rendering the cells incapable of antigen presentation and unresponsive to cytokine stimulation (57)" does not explain the increase of antibody production in E. histolytica infected group. Why could a diminishing antigen presentation generate higher levels of anti-Plasmodium antigens?

Reply: Very true! Not sure why that statement wasn’t caught. The Discussion has been changed significantly. It now reads, “The decrease in macrophage function does not explain the increase in Ab to EBA-175. One possible explanation is that since malaria and E. histolytica…”

Other observations/questions:

• In the title, add "IgG" to Antibody response. Reply: IgG has been added to title (although not all of the co-authors agree this is necessary).

• Check all scientific names of parasite species for correct formatting in italics. (Example Entamoeba histolytica in the Results section in the abstract)

Reply: The scientific name has been checked and are now in italics.

• Please, mention in the background the region where the study was performed. 

Reply: This information was included in the background section of the Abstract. It is also included in the Materials section. 

• It is necessary to describe and discuss the role of MSP1, MPS2, MSP3, and EBA-175 as markers in serological studies.

Reply: This information has been added to the Discussion.

• Considering that coinfection prevalence is relatively low, I consider that it is important to discriminate with colors or point shapes the individuals MAL-IP-, MAL+IP-, MAL-IP+, MAL+IP+ in Figure 1 B-C-D-E

Reply: We thank the Reviewer for the suggestion. However, information in Fig 1B-E is designed to address the question, are Ab prevalence and levels influence by age? Whereas, Fig 2 provides comparisons between individuals infected with malaria alone or co-infected with specific intestinal parasites. Thus, colored dots or symbols are not needed in Fig 1 (and could be confusing to the reader). 

• In page 6 subtitle "Laboratory detection, quantification and speciation of malaria parasites.", I will not use speciation here. I suggest "Diagnosis and quantification of Plasmodium sp. parasites.

Reply: The header has been changed to read: “Laboratory detection of malaria parasites.”

• (Page 14-15) What type of parasite is "Amoeba"? What is the difference between "Amoeba" and E. histolytica? Traditionally, E. histolytica is considered an amoeba too.

Reply: The figure has been revised to read Intestinal Protozoa. Thanks for pointing out the mis-classification. 

• In table 1, to facilitate reading, please remove symbols % and /ul located in cells with data and add to the columns describing the variables.

Reply: The symbols in the data cells have been removed. 

• For consistency, unify parasitemia vs. parasitemia, anemia vs. anemia in the text and plots.

Reply: The British spelling of parasitaemia, anaemia, and haemoglobin have been used through out the MS. 

• (Page 10) change "The major of helminth parasites" to "The most frequent helminth species detected."

Reply: The change was made as suggested. 

• (Table 2) Check all the total numbers for the "Total IP+" column. For example, for protozoans, the sum is 29+19+4 = 48, and it was reported 47

Reply: This has been verified and corrected to 48 in Table 2

• (Page 13) In plot titles Change Ab (Antibody) to IgG 

Reply: We thank the Reviewer for the comment, but decide not to make the change. Our rationale is that by definition, IgG is a class of immunoglobulin found in the blood; whereas, Ab are plasma proteins that bind specifically with an antigen. What was measured was IgG Ab. Since the serological assay measured IgG Ab that were recorded as MFI (median fluorescence intensity), we think the labels on the Y-Axis (Ab levels -MFI) reflect what was done. The Methods section makes it clear that the Ab were of the IgG class. [Note: Serum IgG levels (which implies mg/ml) were not measured.] 

• (Figure 1E) Add, Change from EBA to EBA-175.

Reply: Change has been made.

• Please verify all references formatting (For example, reference 42 is all in capital letters)

Reply: References have been edited as requested by the reviewer.

Review #3: Comments were in the attachment. 

Reply: In revising the MS, all requested changes were made and additional information provided in the text, including information on the BLAST search. The only request we would not fully address is the prevalence of bednet use in the villages. The only information available is that very few children use bednets. Since the slide-positivity rate of 75.6% for P. falciparum, it is unlikely the bednets are having a major influence on the current study. The following information has been added to the MS in the Results section. “To determine if higher Ab levels in children co-infected with P. falciparum and E. histolytica might be due to cross-reactive epitopes, a BLAST search for sequence homology between EBA-175 and E. histolytica proteins was made. No similarities were found using Metablast, and only one hit was found using discontinuous metablast which had a span of only 38 nucleotides (~12 amino acids). Thus, there does not appear to be shared epitopes between these two pathogens that would explain the increase in Ab to EBA-175 in children with co-infections.”

 

Figure for Reviewer #2 confirming an increase in antibody levels with age.

---

## [Decision Letter · Decision Letter 1]

13 Aug 2020

PONE-D-20-12369R1

Full-title: The immunoglobulin G antibody response to malaria merozoite antigens in asymptomatic children co-infected with malaria and intestinal parasites

PLOS ONE

Dear Dr. Ndiabamoh 

Thank you for submitting your manuscript to PLoS ONE. After careful consideration, we felt that your study has the potential to be published if it is revised to address specific topics raised by the reviewers. A major concern was because the authors revised the MS just by deleting part of the text, without rewriting or providing additional analysis as requested. Also, it is essential to clarify in the abstract  that population received mass drug administration, which may explain the lower prevalence of helminths/ protozoans. For your guidance, a copy of the reviewers' comments was included below. 

We look forward to receiving your revised manuscript.

Kind regards,

Luzia Helena Carvalho, Ph.D.

Academic Editor

PLOS ONE

Reviewers' comments:

Reviewer's Responses to Questions

**Comments to the Author**

1. If the authors have adequately addressed your comments raised in a previous round of review and you feel that this manuscript is now acceptable for publication, you may indicate that here to bypass the “Comments to the Author” section, enter your conflict of interest statement in the “Confidential to Editor” section, and submit your "Accept" recommendation.

Reviewer #1: All comments have been addressed

Reviewer #2: (No Response)

Reviewer #3: (No Response)

2. Is the manuscript technically sound, and do the data support the conclusions?

Reviewer #1: Yes

Reviewer #2: Yes

Reviewer #3: Partly

3. Has the statistical analysis been performed appropriately and rigorously? 

Reviewer #1: Yes

Reviewer #2: Yes

Reviewer #3: Yes

4. Have the authors made all data underlying the findings in their manuscript fully available?

Reviewer #1: Yes

Reviewer #2: Yes

Reviewer #3: Yes

5. Is the manuscript presented in an intelligible fashion and written in standard English?

Reviewer #1: Yes

Reviewer #2: Yes

Reviewer #3: No

6. Review Comments to the Author

Reviewer #1: The authors have addressed all the comments.

There are some minor corrections:

Line 36: Please correct the number of children, 244/230 to 244/320

Line 61: Please correct the species, Trichuria trichuria, to Trichuris trichiura

Please use the reference style outlined by the International Committee of Medical Journal Editors (ICMJE)

Reviewer #2: Comments:

1. Check carefully all the italics for species names

2. Is it necessary the sentence in the line 342 in italics?

3. As I previously commented, the discussion about MSP1, MSP2, and MPS3 role as maskers is absent. Please discuss and compare findings with other studies using the same proteins.

4. Please mention in results that hookworms were not detected in the study

5. Please mention in results that no significant difference was observed between the malaria antibodies levels and parasites eggs counts.

6. Although the authors specify two references describing the bead assay, I still believe it is important to report each antigen's concentration in the bead assay. Once the authors explained that these concentrations are not comparable. Additionally, in the references given (old 23 and 24), there is no information about how EBA175 was produced.

7. Please add references showing that eosinophilia is maintained in individuals that had intestinal infections during a longer time even after MDA.

8. The author explains that "The fact that Intestinal parasite (IP) infections was only observed in children >2 years, helped guide separation of the children into seven groups." But that is not e real explanation; please explain what the rationale for that separation is? Why not use 3-4-5-6 groups?

9. Please comment in the abstract that the population received mass drug administration. I consider that that is a crucial point of the investigation and explains the lower prevalence of helminths and protozoans.

10. I think it would be necessary for the manuscript to verify the existence of other articles reporting intestinal parasites in populations under MDA treatment and add that to the discussion.



Reviewer #3: Please see comments in the attached file

7. PLOS authors have the option to publish the peer review history of their article (what does this mean?). If published, this will include your full peer review and any attached files.

Reviewer #1: No

Reviewer #2: No

Reviewer #3: No

---

## [Author Response · Author response to Decision Letter 1]

17 Sep 2020

Response to reviewers comments 

Review Comments to the Author

Reviewer #1: The authors have addressed all the comments.

There are some minor corrections:

Line 36: Please correct the number of children, 244/230 to 244/320

Reply: This correction has been made 

Line 61: Please correct the species, Trichuria trichuria, to Trichuris trichiura

Reply: Trichuria trichuria has been corrected to Trichuris trichiura

Please use the reference style outlined by the International Committee of Medical Journal Editors (ICMJE)

Reply: The referencing has been changed to one of ICMJE editors’ style (U.S. National Library of Medicine style) and edited.

Reviewer #2: Comments:

1. Check carefully all the italics for species names

Reply: We thank the reviewer for this comment, and this has been checked and edited throughout the document. 

2. Is it necessary the sentence in the line 342 in italics?

Reply: We thank the reviewer for this comment. This statement was highlighted to emphasize the statement. The italic has been removed.

3. As I previously commented, the discussion about MSP1, MSP2, and MSP3 role as maskers is absent. Please discuss and compare findings with other studies using the same proteins.

Reply: Please forgive us for not making the change previously, but we are having difficulty understanding the comment in the context of the current study. Clearly, MSP1, MSP2, and MSP3 are considered as vaccine candidates, but, like all antibodies, Ab to MSP1, MSP2 and MSP3 are simply “markers” of exposure. Over the last 10 years, the presence of Ab to merozoite antigens, not only MSP1, MSP2 and MSP3, are considered to serve as markers for infection with malaria (Patel P et al 2017, Suman et al 2010). Likewise, we are confused about the request to “discuss and compare findings with other studies using the same proteins,” since reference to other studies that directly relate to the current study have been made. Since many studies have documented that Ab prevalence and levels increase with age, it seems unnecessary to provide multiple references to document this point. 

In an attempt to respond appropriately to the comments, a paragraph has been added to the Discussion. The paragraph reads: “This study measured the Ab response to 4 antigens present in P. falciparum merozoites. Following schizont rupture, the merozoite surface antigens (MSPs) participate in initial attachment of merozoites to erythrocytes; then erythrocyte-binding antigen EBA-175 aids in binding and induces release of proteins localized in the micronemes that participates in junction formation [Beeson et al. - review]. Thus, the immune system of children living in Etoudi, Cameroon, would be exposed to these antigens simultaneously and repeatedly. Although considered to be vaccine candidates, Ab against these antigens are markers of past or current infection and not necessarily markers of protection [Beeson et al]. Antibodies against any one of the antigens alone is not an index of immunity to malaria. Thus, even if intestinal parasitic pathogens had down-regulated humoral responses to P. falciparum, reduced Ab levels would not necessarily mean the individual had increased susceptibility to malaria disease. Studying these antigens, however, provided a way to assess the impact of intestinal pathogens on humoral immunity to asexual P. falciparum parasites.”

4. Please mention in results that hookworms were not detected in the study

Reply; thank the reviewer for this comment. It has been included accordingly in the text (line 232) 

5. Please mention in results that no significant difference was observed between the malaria antibodies levels and parasites eggs counts.

Reply: Sorry, we humbly beg to differ with the Reviewer and have not complied with the suggestion. Since only 9 subjects had Ascaris lumbricoides infections, the sample size is simply too small to determine if an association exists between Ab levels and egg counts. That is, egg counts ranged only from 100 to 320 eggs per gram of stool; whereas, Ab levels (MFI) had a wide range: (median: 25th, 75th percentile: MSP1 - 2088: 696,1367, 7191; MSP2 - 1778:1507,2750; MSP3 - 6778:434,14,323; EBA-175 10,944:1486, 15058 MFI for these 9 individuals). Using a simple linear regression model, none of the p values were significant (p values = 0.19 to 0.80), demonstrating that none of the associations (regression lines) differed significantly from zero. Although, “technically” there was no statistical association between egg counts and Ab levels, we feel that drawing the conclusion that there was no association is invalid and not good science. 

6. Although the authors specify two references describing the bead assay, I still believe it is important to report each antigen's concentration in the bead assay. Once the authors explained that these concentrations are not comparable. Additionally, in the references given (old 23 and 24), there is no information about how EBA175 was produced.

Reply: Information on EBA-175, along with the other antigens used, was provided in reference 23 (Fouda et al. 2006). Information on the concentrations and expression systems has been added to the Methods section. “Antigens used included MSP142 of the FVO and 3D7 strains expressed in Escherichia coli and coupled at 0.2 µg per million beads, recombinant EBA-175 RII expressed in yeast coupled at 2.5 µg per million beads, recombinant MSP-3 C-terminal region expressed in E. coli coupled at 5 µg per million beads and recombinant MSP-2 (FC27 strain) coupled at 1 µg per million beads. (Fouda et al. 2006, Fodjio et al. 2016).”

7. Please add references showing that eosinophilia is maintained in individuals that had intestinal infections during a longer time even after MDA.

Reply: The only statement in the manuscript concerning eosinophils [line 248] is that … with age … “the prevalence of IP and eosinophilia increased.” The statement simply reports the results. Thus, adding references doesn’t seem appropriate. The length of time eosinophils persist after drug treatment is influenced by many variables, e.g., initial worm burden, number of eosinophils produced by the innate immune response, length of the infection prior to drug treatment, persistence of antigen, frequency of MDA, complete/incomplete cure, etc. In Etoudi, MDA was conducted approximately twice a year. Thus, some individuals became infected between MSD campaigns. Thus, providing references stating how long eosinophilia persists for an extended period after MDA is impossible to provide. 

8. The author explains that "The fact that Intestinal parasite (IP) infections was only observed in children >2 years, helped guide separation of the children into seven groups." But that is not e real explanation; please explain what the rationale for that separation is? Why not use 3-4-5-6 groups?

Reply: We did use age groups 3-4, 5-6-year, etc. Since IP were not detected in children 1-2 years of age, it seemed logical to group children into 2-year bins. There was no other rationale. Grouping the children made it easier to graph the results. The same conclusion would be obtained if the children had been separated into larger sized groups or had not been grouped. 

9. Please comment in the abstract that the population received mass drug administration. I consider that that is a crucial point of the investigation and explains the lower prevalence of helminths and protozoans.

Reply: We thank the reviewer for this comment, and it has been included in the second line of the methods section on the abstract. 

10. I think it would be necessary for the manuscript to verify the existence of other articles reporting intestinal parasites in populations under MDA treatment and add that to the discussion.

Reply: Mass drug administration (MDA) is a common practice in subtropical and tropical countries like Cameroon. The aim is to eliminate intestinal parasitic infections. However, complete eradication is usually not attained as other factors such as hygiene and sanitation, behavior of population and awareness are all require in an integrated manner to avert the problem as these infections may return after a MDA in as early as within four to six months (J.C. Dunn et al., 2019, G. Ortu et al., 2016). Adverse events when not well handled usually lead to rejection of MDA by certain families, thereby serving as reservoir for the parasites to continue to circulate in the community (P. Laurenzo et al., 2019). This information has been added to the Discussion as suggested. Line 385 to 388

Reviewer #3: Please see comments in the attached file

Reply: We thank the Reviewer for his/her helpful suggestions. Clearly, the reviewer is a highly qualified parasitologist. All requests for changes highlighted in the manuscript have been made, except for one. Reviewer #3 originally requested data from the Blast analysis. We provided the following information in the revised manuscript: “No similarities were found using Metablast and only one hit was found using discontinuous metablast which had a span of only 38 nucleotides (~13 amino acids) that had 82% similarity. Thus, there does not appear to be shared epitopes between these two pathogens that would explain the increase in Ab to EBA-175 in children with co-infections.” Reviewer #3 has again requested “Please show the data.” We are not sure what more “data” we can provide, other than to include the nucleotide sequence. So, the sequence has been added to the Discussion.

---

## [Decision Letter · Decision Letter 2]

7 Oct 2020

PONE-D-20-12369R2

Full-title: The immunoglobulin G antibody response to malaria merozoite antigens in asymptomatic children co-infected with malaria and intestinal parasites

PLOS ONE

Dear Dr. Ndiabamoh,

After careful consideration, we have concluded that your manuscript has the potential to be published although some aspects of the manuscript will need to be changed prior to formal acceptance. More specifically, the authors should discuss in much more details the MSP-1/2/3 results comparing with other studies from the same/different areas/individuals. These comments would enrich the manuscript, even considering that the authors could not test the original coinfection effect hypothesis due to the lack of sample size.   

We look forward to receiving your revised manuscript.

Kind regards,

Luzia Helena Carvalho, Ph.D.

Academic Editor

PLOS ONE

Reviewers' comments:

Reviewer's Responses to Questions

**Comments to the Author**

1. If the authors have adequately addressed your comments raised in a previous round of review and you feel that this manuscript is now acceptable for publication, you may indicate that here to bypass the “Comments to the Author” section, enter your conflict of interest statement in the “Confidential to Editor” section, and submit your "Accept" recommendation.

Reviewer #1: All comments have been addressed

Reviewer #2: All comments have been addressed

2. Is the manuscript technically sound, and do the data support the conclusions?

Reviewer #1: Yes

Reviewer #2: Yes

3. Has the statistical analysis been performed appropriately and rigorously? 

Reviewer #1: Yes

Reviewer #2: Yes

4. Have the authors made all data underlying the findings in their manuscript fully available?

Reviewer #1: Yes

Reviewer #2: Yes

5. Is the manuscript presented in an intelligible fashion and written in standard English?

Reviewer #1: Yes

Reviewer #2: Yes

6. Review Comments to the Author

Reviewer #1: The authors have addressed all the comments.

There are some minor corrections:

hook worm should be hookworm.

Page 19 line 397; ref 59 has been written three times.

Reviewer #2: (No Response)

7. PLOS authors have the option to publish the peer review history of their article (what does this mean?). If published, this will include your full peer review and any attached files.

Reviewer #1: No

Reviewer #2: No

---

## [Author Response · Author response to Decision Letter 2]

23 Oct 2020

Response to the Reviewers

Request: “…, the authors should discuss in much more details the MSP-1/2/3 results comparing with other studies from the same/different areas/individuals. These comments would enrich the manuscript, even considering that the authors could not test the original coinfection effect hypothesis due to the lack of sample size,” 

Reply: A paragraph has been inserted into the Discussion that compares the results from this study with those reported previously in Cameroon. Our results are similar to those reported previously; however, a number of variables exist among the studies that make direct comparisons difficult. That is, since different assay techniques (e.g., ELISA vs. multiplex), serum dilutions, and recombinant proteins were used, comparing antibody levels is not possible. Thus, we discuss differences in antibody prevalence in children with age among studies; however, antibody prevalence is influenced by rate of malaria transmission which differed among the studies. The paragraph conclude that many factors influence the antibody response to Plasmodium falciparum, but our results are similar to those reported previously. There are no studies on the influence of E. histolytica on antibody responses to malarial antigens. Thus, we could not include results from additional studies in our Discussion. We sincerely hope the paragraph will “enrich the manuscript.”

In the attachment, Reviewer #2 agreed with many of the changes made during the second revision. The only major comment was that, ”Since the authors could not test the hypothesis about the effect of coinfections in the antibody response due to the low frequency of helminths in the studied population, I was expecting more discussion about the antibody levels against MSP1, MSP2, and MSP3 proteins. I wanted to see other manuscripts studying MSP1, MSP2, and MSP3 because I expected to see if similar levels of antibodies anti MSPs proteins in other populations/age groups, or even see comments about the association with protozoan coinfections. I know that these merozoite proteins are markers of parasite exposure, but at any moment the authors mentioned why those merozoite proteins were chosen over other Pf proteins (sporozoite or even sexual-stage). This type of information is essential, mainly for non-malaria researchers that will read the manuscript.”

Our Reply: The paragraph has been added to the manuscript.

 The Reviewer had two additional suggestions for addition to the manuscript. 

1. The authors should consider adding “The length of time eosinophils persist after drug treatment is influenced by many variables, e.g., initial worm burden, number of eosinophils produced by the innate immune response, length of the infection prior to drug treatment, persistence of antigen, frequency of MDA, complete/incomplete cure” as a complement to the eosinophils discussion. We did not add the statement since our goal was simply to report the results, not to make a general statement about the persistence or life-span of eosinophils. 

2. Our previous reply: Reply: Mass drug administration (MDA) is a common practice in subtropical and tropical countries like Cameroon. The aim is to eliminate intestinal parasitic infections. However, complete eradication is usually not attained as other factors such as hygiene and sanitation, behavior of population and awareness are all require in an integrated manner to avert the problem as these infections may return after a MDA in as early as within four to six months (J.C. Dunn et al., 2019, G. Ortu et al., 2016). Adverse events when not well handled usually lead to rejection of MDA by certain families, thereby serving as reservoir for the parasites to continue to circulate in the community (P. Laurenzo et al., 2019). This information has been added to the Discussion as suggested. Line 385 to 388 

The Reviewer: A =/ Thanks for the answer. Could the authors please add that information as an additional comment to the results in the discussion section?

Reply: It has been added.

---

## [Editor Report · Decision Letter 3]

26 Oct 2020

Full-title: The immunoglobulin G antibody response to malaria merozoite antigens in asymptomatic children co-infected with malaria and intestinal parasites

PONE-D-20-12369R3

Dear Dr. crespo'o mbe-cho ndiabamoh,

We’re pleased to inform you that your manuscript has been judged scientifically suitable for publication and will be formally accepted for publication once it meets all outstanding technical requirements.

Kind regards,

Luzia Helena Carvalho, Ph.D.

Academic Editor

PLOS ONE
---

## [Editor Report · Acceptance letter]

28 Oct 2020

PONE-D-20-12369R3 

The immunoglobulin G antibody response to malaria merozoite antigens in asymptomatic children co-infected with malaria and intestinal parasites 

Dear Dr. Ndiabamoh:

I'm pleased to inform you that your manuscript has been deemed suitable for publication in PLOS ONE. Congratulations! Your manuscript is now with our production department. 

Kind regards, 

on behalf of

Dr. Luzia Helena Carvalho 

Academic Editor

PLOS ONE